# Structural insights into transcription initiation by yeast RNA polymerase I

Yashar Sadian[1], Lucas Tafur[1], Jan Kosinski[1], Arjen J Jakobi[1,2,3], Rene Wetzel[1], Katarzyna Buczak[1], Wim JH Hagen[1], Martin Beck[1] (ID), Carsten Sachse[1] (ID) & Christoph W Müller[1,*] (ID)

## Abstract

In eukaryotic cells, RNA polymerase I (Pol I) synthesizes precursor ribosomal RNA (pre-rRNA) that is subsequently processed into mature rRNA. To initiate transcription, Pol I requires the assembly of a multi-subunit pre-initiation complex (PIC) at the ribosomal RNA promoter. In yeast, the minimal PIC includes Pol I, the transcription factor Rrn3, and Core Factor (CF) composed of subunits Rrn6, Rrn7, and Rrn11. Here, we present the cryo-EM structure of the 18-subunit yeast Pol I PIC bound to a transcription scaffold. The cryo-EM map reveals an unexpected arrangement of the DNA and CF subunits relative to Pol I. The upstream DNA is positioned differently than in any previous structures of the Pol II PIC. Furthermore, the TFIIB-related subunit Rrn7 also occupies a different location compared to the Pol II PIC although it uses similar interfaces as TFIIB to contact DNA. Our results show that although general features of eukaryotic transcription initiation are conserved, Pol I and Pol II use them differently in their respective transcription initiation complexes.

**Keywords** Core Factor; electron cryo-microscopy; Pol I pre-initiation complex; ribosomal RNA; transcription

**Subject Categories** Structural Biology; Transcription

**The EMBO Journal (2017) 36: 2698–2709**

See also: **L Jochem et al** (September 2017)

## Introduction

Biogenesis of ribosomes is dependent on the high transcriptional activity of eukaryotic RNA polymerase I (Pol I). Pol I contributes more than 60% to the total cellular transcription activity of the cell and accordingly requires high transcription initiation rates (Schneider, 2012). During transcription initiation, Pol I, similar to other eukaryotic RNA polymerases, assembles together with its specific transcription factors on the ribosomal RNA promoter to

form the pre-initiation complex (PIC; Murakami et al, 2013). In sequential steps, the PIC transitions from a closed complex (CC), in which the DNA is still double-stranded, to an open complex (OC), in which the double-stranded DNA in the vicinity of the transcription start site is melted. Finally, an initially transcribing complex (ITC) forms when short RNA is synthesized just before RNA polymerase enters the elongation phase (Plaschka et al, 2015; Sainsbury et al, 2015; He et al, 2016).

For initiating rRNA synthesis, yeast Pol I uses a unique set of transcription factors: Upstream Activating Factor (UAF), TATA-binding protein (TBP), Core Factor (CF), and Rrn3 (Keener et al, 1998). UAF and TBP bind to the upstream promoter and have only a stimulatory effect in vitro (Keener et al, 1998), while CF, composed of Rrn7, Rrn6, and Rrn11, binds to the core promoter element (−38 to +5; Kulkens et al, 1991; Meier & Thoma, 2005) and is essential for the basal level of transcription (Keener et al, 1998). Rrn7 is a TFIIB-like factor, has a similar domain structure as TFIIB and Brf1, comprising two central cyclin domains preceded by an N-terminal Zn-ribbon, B-reader, and B-linker, and had been predicted to be followed by a Pol I-specific C-terminal extension absent in TFIIB (Knutson & Hahn, 2011; Naidu et al, 2011). Rrn6 contains a β-propeller followed by a helical domain, while Rrn11 comprises a central tetratricopeptide repeat (TPR) domain preceded by an N-terminal extension (Knutson et al, 2014). Rrn3 stabilizes the monomeric form of Pol I and interacts with CF to position Pol I on the rDNA promoter thereby promoting the formation of a productive Pol I PIC (Bedwell et al, 2012). Altogether, the minimal Pol I PIC (comprised of Pol I, Rrn3, and CF) is equipped with the necessary elements for promoter-dependent transcription, namely start site selection, DNA melting, and promoter escape.

Even though transcription initiation in bacterial Pol or eukaryotic Pol II has been extensively studied (Campbell et al, 2008; Murakami et al, 2013, 2015; Feklistov et al, 2014; Ruff et al, 2015; He et al, 2016; Plaschka et al, 2016), so far initiation steps in the Pol I system have not been structurally characterized in detail. While Pol II requires the assembly of several general transcription factors for transcription initiation, bacterial Pol only requires the action of sigma factor (Feklistov et al, 2014). Recent studies have proposed models for the minimal Pol I PIC based on Pol II PIC, biochemical

1   European Molecular Biology Laboratory (EMBL), Structural and Computational Biology Unit, Heidelberg, Germany
2   European Molecular Biology Laboratory (EMBL), Hamburg Unit, Hamburg, Germany
3   The Hamburg Centre for Ultrafast Imaging (CUI), Hamburg, Germany
    *Corresponding author. Tel: +49 6221 387 8320; E-mail: cmueller@embl.de

data, and homology between Rrn7 and TFIIB (Vannini & Cramer, 2012; Knutson *et al*, 2014; Hoffmann *et al*, 2016). In addition, crystal structures of yeast Pol I (Engel *et al*, 2013; Fernandez-Tornero *et al*, 2013) and more recently structures of the Pol I-Rrn3 complex (Engel *et al*, 2016; Pilsl *et al*, 2016) and of transcribing Pol I (Neyer *et al*, 2016; Tafur *et al*, 2016) have become available while structural information on the CF is still lacking.

Here, we present the model of a yeast Pol I PIC in complex with CF, Rrn3, and a transcription scaffold resolved to 4.4 Å using single-particle cryo-electron microscopy (cryo-EM). The model shows several differences to the Pol II PIC (Plaschka *et al*, 2016) and to previously proposed models of the Pol I PIC (Knutson *et al*, 2014; Hoffmann *et al*, 2016). CF directly contacts promoter DNA upstream of Pol I through Rrn7 and a helical bundle that together bend the DNA and change its direction compared to the Pol II PIC (Plaschka *et al*, 2016). The Rrn11 TPR domain is positioned in close proximity to the Pol I A135 protrusion, while the Rrn6 β-propeller is located at the opposite side of the DNA with respect to Rrn7, possibly serving as a protein–protein interaction platform for TBP. Our results show that the molecular architecture of the Pol I PIC is structurally different from previously observed bacterial or eukaryotic Pol II PICs (Feklistov *et al*, 2014; Plaschka *et al*, 2016).

# Results

### Cryo-EM structure of the Pol I PIC

We stepwise assembled the minimal Pol I PIC by adding recombinant CF, Rrn3, and natively purified, transcriptionally active Pol I to a 70 base pair (bp) transcription scaffold (−50 to +20) containing the core rDNA promoter (−38 to +5) with a 15-nucleotide (nt) mismatch and a 10-nt RNA (Tafur *et al*, 2016). A homogenous sample containing the 18-subunit protein complex with a molecular mass of ~900 kDa was obtained using dialysis from high to low salt, whereas the complex did not form in the absence of DNA (Appendix Fig S1A–D, see Materials and Methods). An initial negative stain reconstruction revealed the overall position of CF relative to Pol I and provided a starting model for the analysis of the cryo-EM data. We collected 4,235 movie frames on a FEI Titan Krios equipped with a Gatan K2 Summit direct electron detector and processed them using RELION (Scheres, 2012). Initial analysis revealed that most of the complex dissociated and the majority of the particles corresponded to Pol I bound to DNA (Tafur *et al*, 2016). Extensive particle classification allowed excluding particles containing only Pol I or Pol I-Rrn3 and improved the density corresponding to CF (Appendix Fig S2), leading to 38,589 Pol I initiation complex particles that were refined to an average resolution of 4.4 Å (Figs 1 and EV1A). Focused refinements on CF and Pol I-Rrn3 resulted in reconstructions with enhanced details corresponding to Pol I and the upstream DNA, but had limited effect on the CF density. We attribute this to a combination of intrinsic flexibility of the CF bound to upstream DNA and uneven angular distribution of the particles (Fig EV1B), both factors limiting the resolution of the cryo-EM maps, in particular in peripheral regions of CF (Fig EV1C).

In the reconstructions resulting from the focused refinements, higher resolution features are only observed in the core region of Pol I and CF, the former showing increased details including

side-chain densities (Fig EV1C and D). Although Rrn3 is present in the reconstruction, it shows lower resolution compared to all other proteins (Fig EV1E). Throughout the map, secondary structure elements were clearly visible, allowing fitting of homology models of the Rrn7 cyclin domains and the Rrn6 β-propeller. To aid in modeling the remaining densities, we performed cross-linking mass spectrometry of purified Pol I PIC using di-succinimidyl-suberate (DSS) cross-linker. We obtained 124 unique inter-links and 194 unique intra-links with the LD (linear-discriminant) confidence score at least 23 as calculated by xQuest (Leitner *et al*, 2014). The appropriateness of the score threshold was validated using the Pol I crystal structure (PDB ID: 4c3i), for which 113 out of 116 cross-links (97.4%) mapped to the structure satisfied the distance threshold of 30 Å. With the aid of the cross-linking data and the identification of macromolecular folds guided by the cryo-EM density (Materials and Methods), we could assign most of the Rrn11 helices, which resulted in a topological model of the Pol I initiation complex (Fig 1A). The model includes the DNA-bound Pol I-Rrn3 complex (Engel *et al*, 2016; Pilsl *et al*, 2016), downstream and upstream DNA, the Rrn7 N-terminal Zn-ribbon (B-linker and B-reader are not resolved), the Rrn7 cyclin domains, most of the Rrn11 TPR domain, and the Rrn6 β-propeller domain (Fig 1B). In addition, we observed several helical densities throughout the CF-upstream DNA region, including the DNA binding helical bundle (DBHB), which we were unable to unambiguously assign to a specific CF subunit.

### Unexpected DNA position in the Pol I PIC

The cryo-EM density reveals almost the entire path for the downstream and upstream DNA, although clear density for the single-stranded open DNA region is missing. We did not observe any density for the RNA, which was either lost during sample preparation or cleaved by the catalytic domain of the A12.2 subunit of Pol I similar to the human Pol II ITC (+TFIIS; He *et al*, 2016; Fig 2A). The upstream promoter DNA is bound by CF approximately from base pair (bp) ~−40 to ~−16 (Fig 2B) immediately upstream of the region bound by Pol I. The position of CF on the promoter DNA agrees with previously published data showing that CF interacts with the "core" promoter region (mapped to bp −38 to +5; Kulkens *et al*, 1991; Meier & Thoma, 2005).

Based on the similarity between Pol I and Pol II and their transcription factors, the DNA path in the Pol I and Pol II PICs was expected to be similar (Vannini & Cramer, 2012; Knutson *et al*, 2014; Hoffmann *et al*, 2016). Surprisingly, in the Pol I PIC, the upstream DNA duplex follows a remarkably different path (Fig 2C). In the Pol II PIC, the upstream DNA is bound outside of the cleft by several transcription factors (Murakami *et al*, 2015; He *et al*, 2016; Plaschka *et al*, 2016), whereas in the Pol I PIC, the upstream DNA is positioned closer to the wall and protrusion domains. Moreover, upstream DNA in the Pol I elongation complex (EC) re-anneals close to the protrusion domain and the positive helix (Tafur *et al*, 2016), while in the Pol I PIC, the upstream DNA is even closer to the wall and to the A135 "wedge" (residues 813–819; Barnes *et al*, 2015; Fig EV2A). The upstream DNA also comes closer to another loop of A135 (residues 890–897) that extends from the A135 wall that probably stabilizes the DNA position (Fig EV2B). Consequently, changing the DNA conformation from initiation to elongation requires only a small shift of the upstream DNA (Fig EV2C).

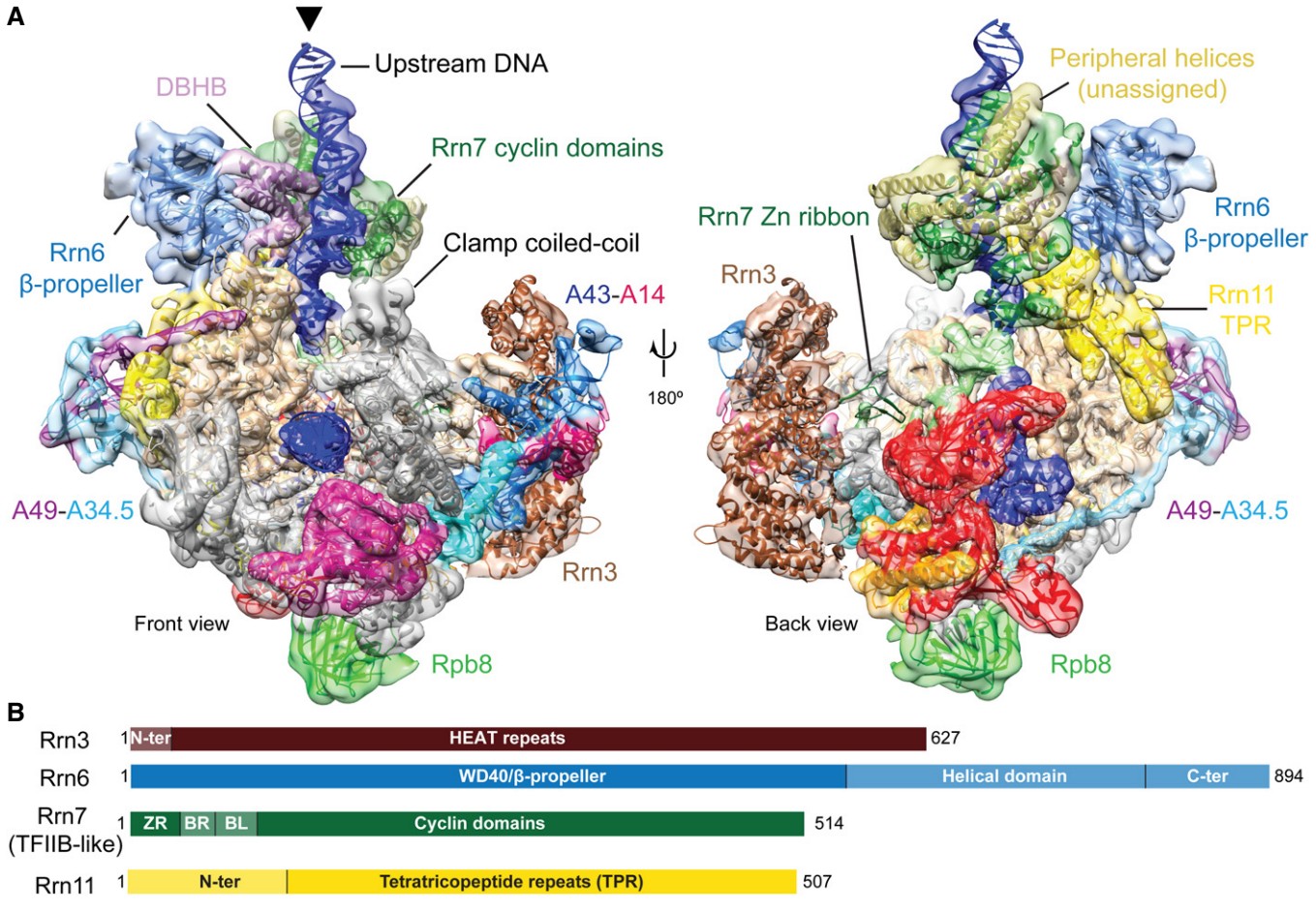

**Figure 1. Cryo-EM structure of the Pol I PIC.**

A  Density for the globally refined Pol I PIC cryo-EM map is shown in two views colored according to the subunits, as in (B) and in Fernandez-Tornero et al (2013). For Rrn3, density of the Pol I-Rrn3 focused refinement at a lower density threshold than Pol I is displayed. Unassigned helical densities in CF are shown in gold color. The cryo-EM densities have been filtered according to the local resolution. The threshold of the cryo-EM map was chosen to clearly visualize secondary structure elements. The density for the Rrn7 Zn-ribbon is not visible at this threshold but is depicted in Fig 3A and B.

B  Schematic representation for Rrn3 and the CF subunits Rrn6, Rrn7, and Rrn11. Lighter colors denote major regions not observed in the density.

In contrast, the downstream DNA in the Pol I PIC is in a similar position to previous DNA-bound structures of Pol I (Neyer et al, 2016; Tafur et al, 2016) and Pol II (Gnatt et al, 2001; Fig 2C) demonstrating that downstream DNA is retained inside the DNA binding cleft during the transition from initiation to elongation. These findings also agree with previous structures showing that Pol I and Pol II can position downstream DNA correctly in the absence of transcription factors and RNA (Cheung et al, 2011; Tafur et al, 2016).

**Rrn7 and the DNA binding helical bundle position the DNA**

The cryo-EM structure of the Pol I PIC also reveals a unique arrangement of the Rrn7 subunit, which based on its similarity to TFIIB was expected to bind Pol I and promoter DNA at a position equivalent to Pol II (Vannini & Cramer, 2012; Knutson et al, 2014; Hoffmann et al, 2016). The Zn-ribbon domain, although resolved weakly, indeed occupies a similar position as the TFIIB B-ribbon (Fig 3A). However, the position of the Rrn7 cyclin domains relative to Pol I is very different. In the Pol II-TFIIB crystal structures (Kostrewa et al, 2009;

Liu et al, 2010; Sainsbury et al, 2013) and in the Pol II PIC (Plaschka et al, 2016), the N-terminal cyclin domain is located in close proximity to the Pol II wall and the C-terminal cyclin domain of TFIIB is positioned next to the Rpb2 protrusion and Rpb12. In contrast, in the Pol I PIC, both Rrn7 cyclin domains are located farther away from Pol I and do not directly contact Pol I. Direct interactions with Pol I are precluded in the Pol I initiation complex since the corresponding position of the N-terminal cyclin domain near the wall in Pol II is occupied by upstream DNA in our model (Fig 3B).

TFIIB-like factors including TFIIB and Brf2 (a TFIIB-like factor present only in vertebrates) bind DNA utilizing the interfaces formed by helix α7 and a turn between helices α4 and α5 (Sainsbury et al, 2013; Gouge et al, 2015). Despite its different position, the N-terminal cyclin domain of Rrn7 also uses this interface (Fig 3C). Interestingly, the exact DNA elements bound by Rrn7 and TFIIB/Brf2 are different. TFIIB and Brf2 bind the minor groove through α4-α5 and the major groove by α7. In the Pol I PIC, however, the corresponding Rrn7 regions appear to bind the phosphate backbone (Fig 3C and see also Fig 2A). Furthermore, the C-terminal cyclin

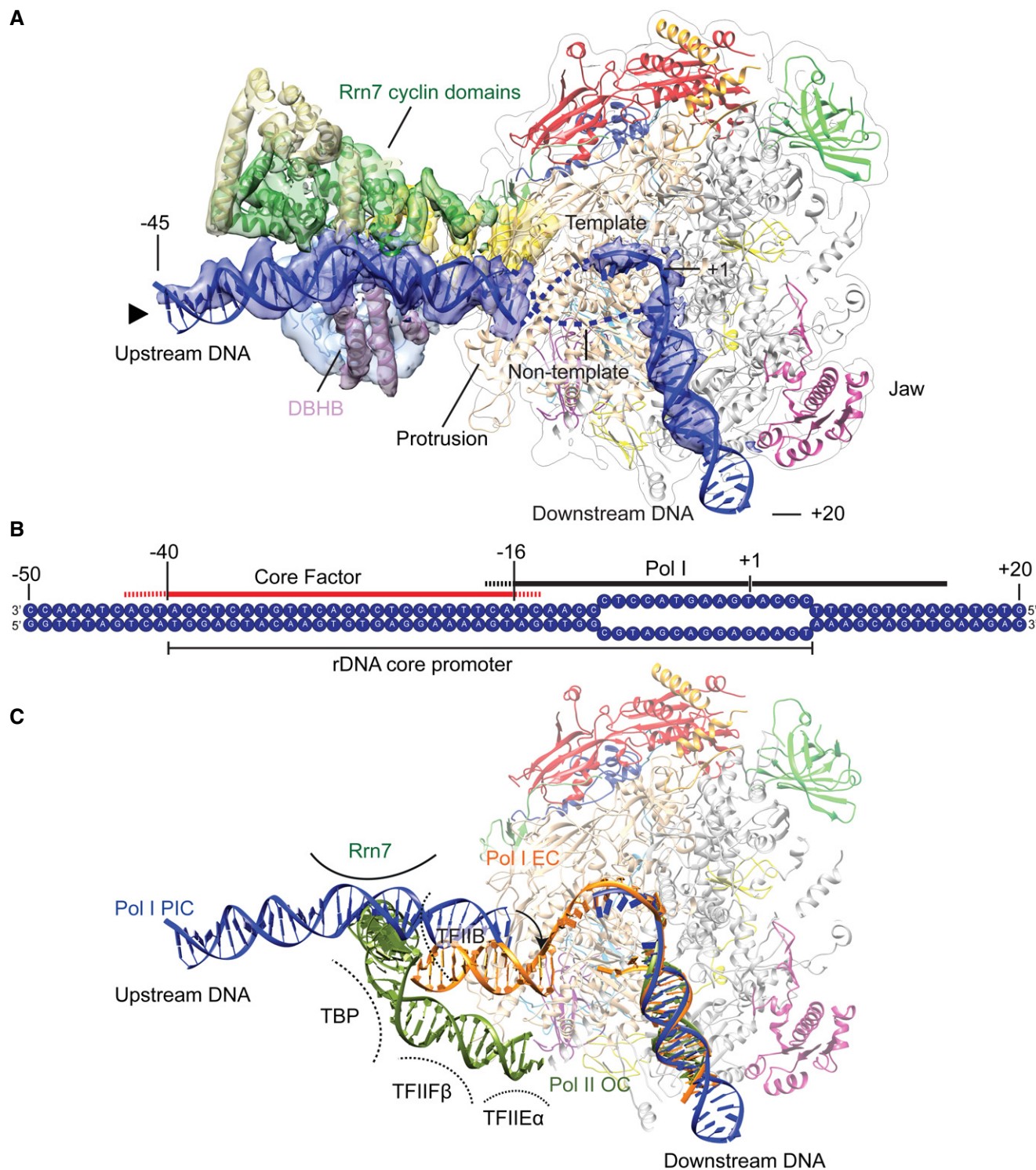

**Figure 2.  DNA conformation and binding in the Pol I PIC.**

A    The segmented density of the downstream (from the Pol I-Rrn3 focused refinement) and upstream DNA (from the CF-focused refinement) is shown in blue. The DNA open region is unresolved. The Rrn7 cyclin domains (green) and the DBHB (medium purple) contact DNA on opposite sides. The cryo-EM densities depicted have been filtered according to the local resolution. The black triangle next to the upstream DNA indicates the perspective of view with respect to Fig 1A.

B    Schematic representation of the DNA scaffold. Regions that come into close contact with CF (red) and Pol I (black) are indicated. Dashed lines for CF and Pol I indicate uncertainty in the boundaries due to the unknown exact sequence register of the DNA.

C    Comparison of the upstream DNA path of the Pol I PIC (blue), Pol I elongation complex (Pol I EC, PDB ID 5m5x; orange), and Pol II open complex (Pol II OC; dark green). Transcription factor interaction surfaces that could influence the position of the upstream DNA are schematically depicted based on PDB ID 5fyw.

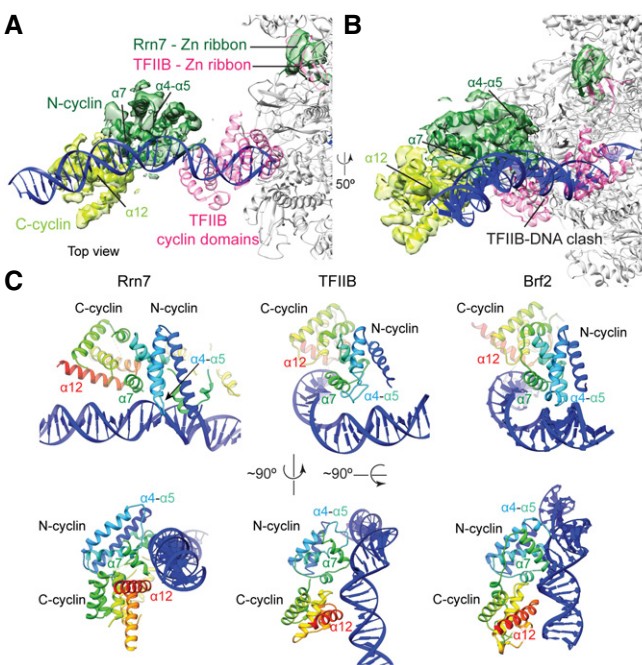

**Figure 3. Rrn7 position and its interaction with DNA.**

A   Density of the N-terminal cyclin domain (N-cyclin) and Zn-ribbon is colored in dark green and of the C-terminal cyclin domain (C-cyclin) in light green. The TFIIB cyclin domains and Zn-ribbon are colored in pink. The position of TFIIB is closer to Pol II than Rrn7 to Pol I, but the Zn-ribbon is in a similar position near the dock domain. The cryo-EM density of the CF-focused refinement filtered according to the local resolution is superimposed, except for the Rrn7 Zn-ribbon, in which the global map is used.

B   Same view as in panel (A) rotated 50° to highlight the overlapping position of the TFIIB cyclin domains and the upstream DNA in our model.

C   The Rrn7 cyclin domains use the same interface for DNA binding as TFIIB and Brf2 (top row) and show a similar relative domain organization (bottom row).

domain of Rrn7 is not positioned as closely to the DNA as in Pol II, where the C-terminal cyclin domain of TFIIB contacts DNA upstream of the TATA box (Plaschka *et al*, 2016). The observed differences could be a result of DNA bending by TBP in the Pol II PIC, which brings the DNA in close proximity to the C-terminal cyclin domain (Murakami *et al*, 2013, 2015; Plaschka *et al*, 2015, 2016; He *et al*, 2016). Nevertheless, the overall conformation of the C-terminal cyclin domain relative to the N-terminal cyclin domain is conserved between TFIIB-like factors (Fig 3C).

We also observe clear helical densities reaching toward the DNA and the Rrn7 cyclin domains from the opposite side of the DNA (Fig 2A). This DNA binding helical bundle (DBHB) does not show clear connectivity to either Rrn6 or Rrn11, which precludes unambiguous assignment to a specific subunit. The DBHB contacts the DNA backbone opposite to the minor groove that is close to the Rrn7 cyclin domains. Together, Rrn7 and the DBHB promote bending of the DNA by ~35° around position ~−30 and likely contribute to maintaining the upstream end of the transcription bubble next to the protrusion.

Mapping of the electrostatic potential onto the molecular surface of the Pol I PIC (Fig EV3) shows that, in addition to direct DNA contacts formed primarily by the Rrn7 N-terminal cyclin domain and the DBHB, other regions also contribute to DNA binding.

Positive patches of the A135 protrusion, the Rrn11 TPR domain, and both Rrn7 cyclin domains form a continuous positively charged binding surface along the DNA path. Although the exact sequence register in the Rrn11 TPR domain in our model is uncertain, the sequence for the helical densities in the TPR domain could be approximately assigned (see Materials and Methods) and we only expect minor sequence register shifts. This allows calculating an approximate electrostatic potential, which is not significantly affected by minor sequence register shifts. Interestingly, the C-terminal cyclin domain of Rrn7 also exposes a positive patch mainly formed by its last helix α12 toward upstream DNA (Figs 3 and EV3), which is likely contributing to stabilize this DNA conformation by electrostatic attraction without directly contacting the DNA.

### Rrn6 and Rrn11 are located in an unexpected position

Early biochemical experiments have shown an interaction between the C-terminus of Rrn6 with Rrn3 (Peyroche *et al*, 2000) and the N-terminus of Rrn6 to TBP (Steffan *et al*, 1996). Therefore, in order to satisfy both, cross-linking and biochemical data, the previously proposed models placed Rrn6 and Rrn11 in a "canyon" between models of Rrn7-DNA and Pol I-Rrn3 (Knutson *et al*, 2014; Hoffmann *et al*, 2016). In the cryo-EM structure of the Pol I PIC, the β-propeller domain of Rrn6 is located at the upstream DNA end in a position where it is more likely to engage in interactions with TBP- and UAF, but not Rrn3 (Fig 4A). The helical domain of Rrn6, as well as the N- and C-terminal regions of Rrn6, could not be located unambiguously. However, we observe helical densities around the cyclin domains of Rrn7 that could represent the helical domain of Rrn6, which cross-links to the cyclin domains according to our cross-linking data (Fig EV4A and B) and previously published data (Knutson *et al*, 2014). The C-terminal half of Rrn11, which encompasses a consensus TPR domain, bridges the Rrn6 β-propeller and the Rrn7 cyclin domains. We were able to assign nine TPR helices at the C-terminal end of Rrn11 into the cryo-EM density (Fig EV4C), while the remaining five helices (according to secondary structure predictions) could not be unambiguously located in the density. The Rrn11 TPR domain is positioned close to Pol I and displaces a loop in the Pol I protrusion (A135 residues 110–119) that otherwise would sterically clash (Fig 4B). Apart from the movement of this loop, Pol I retains essentially the same conformation as seen in the Pol I open complex (OC; PDB ID: 5m5w, Tafur *et al*, 2016), where Pol I has an open cleft, contains the C-terminus of A12.2 in the active site, and where the bridge helix is not fully folded (Fig EV5A–C). The position of Rrn6 and Rrn11 in our model deviates from the previous model of the Pol I PIC, which was based on cross-links of isolated CF and homology between Rrn7 and TFIIB (Knutson *et al*, 2014). We explain this discrepancy by the fact that CF might change its conformation in the presence of DNA and/or Pol I. Indeed, in the Pol I PIC, the DNA makes several contacts to CF and plausibly influences strongly its conformation.

## Discussion

Our Pol I PIC model shows major differences to the Pol II system and to previously proposed Pol I PIC models (Knutson *et al*, 2014; Hoffmann *et al*, 2016). Most prominently, upstream DNA is

**Figure 4. Rrn6 and Rrn11 positions in the Pol I PIC.**

A   The Rrn6 β-propeller is positioned close to the upstream end of the DNA scaffold.

B   The Rrn11 TPR domain is positioned close to the protrusion domain displacing a wall loop formed by A135 residues 110–119. The position of this loop (orange) in the Pol I OC (PDB ID 5m5w) is indicated. The starting and ending residues of the Rrn11 TPR domain are shown. The cryo-EM density of the CF-focused refinement filtered according to the local resolution is superimposed.

positioned differently through interactions with the Rrn7 cyclin domains and the DBHB, which appear to maintain the upstream DNA next to the Pol I protrusion and wall in a position close to that observed for the EC (Tafur *et al*, 2016). This could represent a functional adaptation of Pol I toward a more efficient transcription initiation mechanism where the transition from initiation to elongation is facilitated, thereby increasing the efficiency of promoter escape (Schneider, 2012). Moreover, the cyclin domains of Rrn7 are positioned differently than TFIIB, while the Zn-ribbon is present in a conserved position next to the RNA exit channel. This suggests that the N-terminal region of Rrn7 might function in a similar manner to TFIIB, as proposed previously (Knutson & Hahn, 2013), but that the cyclin domains could have different or additional functions compared to the Pol II system although

they use similar interfaces as TFIIB and Brf2 to interact with promoter DNA.

Our work helps in understanding the Pol I transcription cycle (Fig 5). Before engaging in the cycle, Pol I exists as a dimer (inactive) and a monomer (pre-active) in solution (Pilsl *et al*, 2016) and *in vivo* (Torreira *et al*, 2017). Binding of Rrn3 appears to stabilize the monomeric conformation of Pol I where the cleft is open, the BH partially unfolded, and the A12.2 C-terminal domain present in the active site (Engel *et al*, 2016; Pilsl *et al*, 2016). The Pol I-Rrn3 complex then binds to CF, positioning Pol I for accurate transcription initiation. Our Pol I complex in the presence of Rrn3, CF, and an open DNA scaffold resembles this pre-active conformation as it also exhibits an open cleft, partially unfolded BH, the A12.2 C-terminal domain is in the active site and a poorly resolved A49 tWH domain. Moreover, the template DNA strand in the active site is tilted with respect to the EC (Fig 2C). Transition from our Pol I PIC model to an actively transcribing Pol I EC requires several conformational changes, including displacement of the A12.2 C-terminal domain from the active site, BH folding, clamp/cleft closing, and repositioning of the template DNA.

After our work was submitted, a study describing the crystal structure of CF and cryo-EM structures of Pol I in complex with CF and Rrn3 and of a Pol I ITC with Pol I, CF, and Rrn3 bound to a DNA–RNA transcription scaffold was published (Engel *et al*, 2017). The overall architectures of our model and the Pol I ITC are very similar (Fig EV5D and E). Although the position of the open DNA region is slightly different in the transcription scaffolds, the position of Pol I and CF relative to the DNA is essentially the same confirming that the binding of CF to promoter DNA is highly specific. Moreover, CF maintains the proposed interaction regions with Pol I (Fig EV5D). In addition, Rrn3 is bound to Pol I in similar positions but in our model the C-terminal region is slightly rotated away from the AC40-AC19 binding surface (Fig EV5F). Overall, our CF model also agrees well with the CF crystal structure despite the fact that the cryo-EM map used for model building only extended to 4.4 Å resolution (Appendix Fig S3). A detailed comparison between our CF cryo-EM model and the CF crystal structure is depicted in Appendix Fig S4. Accordingly, the DBHB that was unassigned in our cryo-EM model corresponds to the N-terminal helices of Rrn11, while the peripheral helices (also unassigned in our model) belong to the headlock domain of Rrn6.

Despite their overall similarities, the two Pol I initiation complexes represent different functional states. While our Pol I PIC is in a pre-active conformation exhibiting an open cleft, a partially unfolded BH, and the A12.2 C-terminal domain in the active site, Engel and colleagues report a Pol I ITC in a transcriptionally active conformation where the BH is folded, the cleft has narrowed down, and the A12.2 C-terminal domain is excluded from the active site (Fig EV5E). Our Pol I PIC presumably corresponds to a pre-active Pol I OC that subsequently transitions into a Pol I ITC and finally into a Pol I EC. The observed conformational changes between the Pol I PIC described here and the Pol I ITC correspond to the differences observed between the Pol I OC (Tafur *et al*, 2016) and the Pol I EC (Neyer *et al*, 2016; Tafur *et al*, 2016) in the absence of CF. Like in the Pol I OC in the absence of CF (Tafur *et al*, 2016), the partial cleft closing and the presence of the A12.2 C-terminal domain in the Pol I PIC suggests an intermediate situation where the Pol I

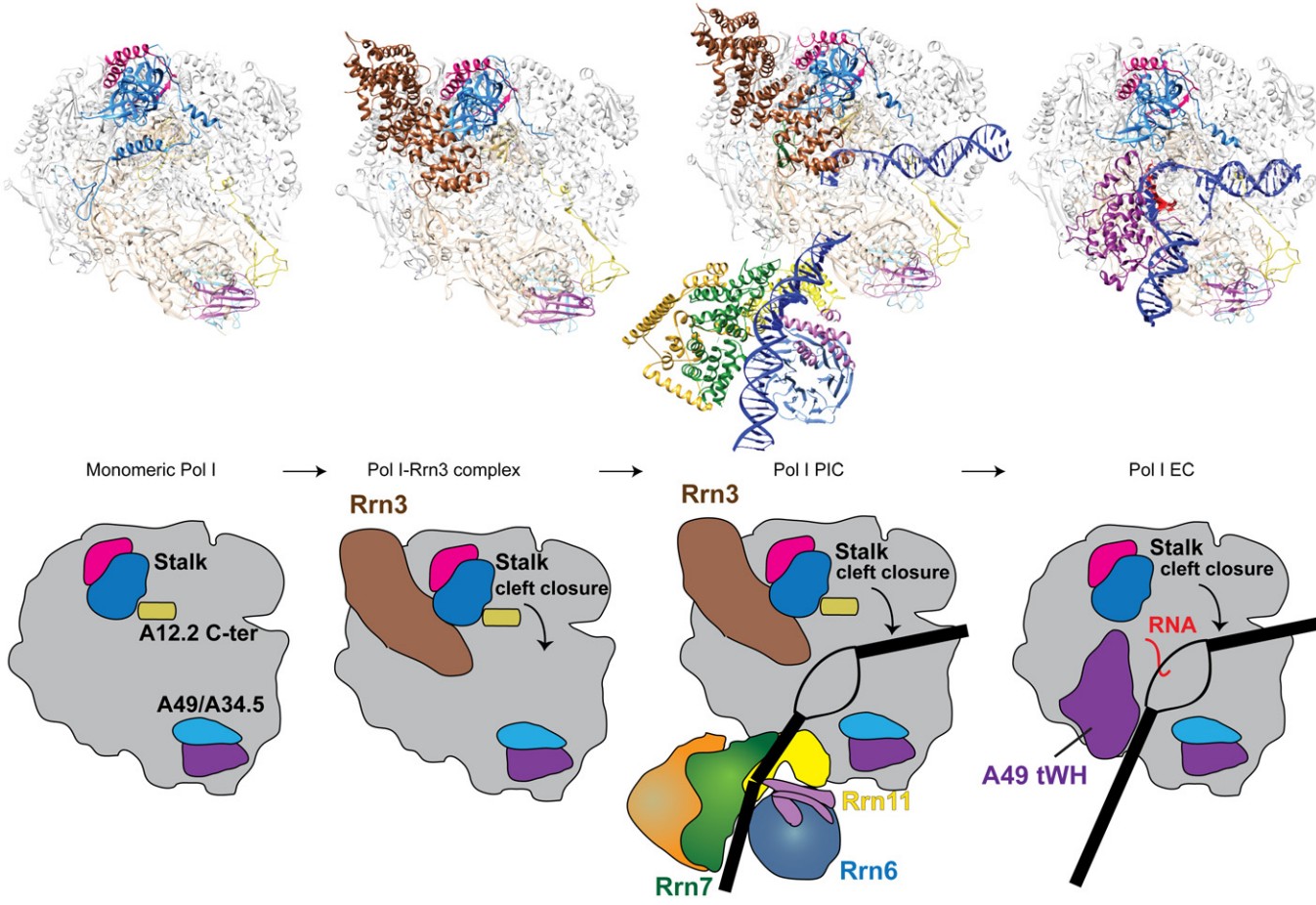

**Figure 5.  Transition from initiation to elongation in Pol I.**

Pol I monomers bind to Rrn3, to form an initiation-competent complex. This complex then associates with CF bound to DNA to form the PIC. The N-terminal region of Rrn7 is close to the active site (Zn-ribbon, B-reader, and B-linker), the Pol I bridge helix (BH) is almost folded, and the catalytic A12.2 C-terminal domain is in the active site. Pol I escape to elongation presumably involves a similar mechanism as in Pol II, where the growing RNA transcript displaces the Rrn7 B-reader and B-linker. Further changes in Pol I (BH folding, clamp closing, exclusion of the A12.2 C from the active site and movement of the upstream DNA) destabilize the late PIC, while the A49 tWH promotes escape to elongation by maintaining the clamp closed and the upstream DNA close to the protrusion positive helix. Molecular models are depicted above the cartoon representation.

PIC further transitions toward an ITC and finally an EC. Conceivably, stable binding of the DNA–RNA hybrid could induce the required conformational changes to form the Pol I active center. Alternatively, our observed Pol I complex may also constitute an abortive intermediate in which RNA has been cleaved due to the intrinsic RNase activity of the A12.2 C-terminal domain. However, structurally distinguishing whether the observed Pol I complex will further transition toward transcription elongation or transition toward abortion might not be possible.

In Pol II, the OC closely resembles the ITC, whereas in Pol I apparently this transition involves several conformational changes. Transcription initiation intermediates in Pol I might deviate from Pol II because of the different requirements for productive initiation, their strikingly distinct architectures, and the presence of Pol I-specific features like the constitutive presence of Pol I-specific TFIIS-, TFIIF-, and TFIIE-like factors. Our study shows that transcription initiation in the Pol I system is topologically different than in Pol II and suggests that although general mechanisms appear to be conserved, Pol I might have structurally evolved to optimize

efficient transcription of rDNA. Further studies aimed at determining the conformation of the closed complex, as well as the complete PIC including UAF and TBP, are necessary to better understand the similarities and differences in transcription initiation between the different eukaryotic RNA polymerases.

## Materials and Methods

### Protein purification and complex assembly

Pol I was purified from *S. cerevisiae* strain SC1613 modified with a C-terminal TAP tag on AC40 as previously described (Fernandez-Tornero *et al*, 2013). Rrn3 was expressed in BL21(DE3)Star pRARE cells using pRSF vector (Novagen) in auto-induction (ZY) medium. The temperature was shifted to 20°C when cultures reached $OD_{600 \, nm}$ 1.2. Cells were harvested after an overnight incubation. The cell pellet was lysed in a buffer containing 1 M NaCl, 50 mM Tris–HCl pH 8, 20% glycerol, 10 mM $MgCl_2$, DNase, and protease

inhibitor cocktail (Roche). The lysate was centrifuged at 48,000 $g$ for 1 h, mixed with Ni-NTA beads (QIAGEN), and incubated for 1 h at 4°C. The beads were washed and then eluted with a buffer containing 100 mM NaCl, 50 mM Tris–HCl pH 8, and 150 mM imidazole. The His-tagged protein was incubated with tobacco etch virus (TEV) protease at 4°C overnight. After His-tag cleavage, the protein was further purified using a Mono-Q column (QIAGEN) and Superdex 200 (GE Healthcare) in 150 mM NaCl, 50 mM Tris–HCl pH 8, and 10 mM DTT. Core Factor (CF) was expressed in BL21 (DE3)Star pRARE cells using pRSF-Duet and pCDF-Duet vector (Novagen) in TB medium. Expression was induced using 0.7 mM IPTG at $OD_{600\ nm}$ 0.7. Cells were then harvested after an overnight incubation at 25°C. Cells were lysed in a buffer containing 0.5 M NaCl, 50 mM Tris–HCl pH 8, 20% glycerol, 10 mM $MgCl_2$, 1 μM $ZnCl_2$, DNase, and protease inhibitor cocktail (Roche). The lysate was cleared with centrifugation and incubated with Ni-NTA for 1 h at 4°C. The beads were washed, and the protein was eluted with a buffer containing 100 mM NaCl, 50 mM Tris–HCl pH8, 10% glycerol and 10 mM $MgCl_2$, 1 μM $ZnCl_2$, and 150 mM imidazole. Tobacco etch virus (TEV) protease was mixed with the eluent and incubated overnight at 4°C. The TEV-cleaved protein was loaded on heparin Sepharose to separate CF using a gradient from 0.1 to 1 M NaCl. Fractions containing CF were further clarified using Superdex 200 column (GE Healthcare) in buffer containing 150 mM NaCl, 50 mM Tris–HCl pH 8, and 10 mM DTT.

## PIC preparation

1.5 μM of Pol I and 3 μM of Rrn3 were mixed and incubated at 4°C for 5 h. Meanwhile, a transcription scaffold containing the core promoter sequence (−50 to +20; Template, 5′-GTCTTCAACTG CTTTCGCATGAAGTACCTCCCAACTACTTTTCCTCACACTTGTACT CCATGACTAAACC-3′; non-template, 5′-GGTTTAGTCATGGAGTA CA AGTGTGAGGAAAAGTAGTTGGCGTAGCAGGAGAAGTAAAGCA GTTGAAGAC-3′, and a 15-nt mismatch region with a 10-nt RNA 5′-GAGGUACUUC-3′ was prepared by heating the template and non-template strands at 95°C for 3 min and cooling down to 25°C in 1 h. Equimolar amount of RNA was then added and a gradient from 45 to 4°C was used to anneal it to the double-stranded DNA. The transcription scaffold was incubated with the Pol I-Rrn3 sample for 1 h at 4°C. Subsequently, 1.5 μM of CF was added in a buffer with 300 mM potassium acetate, 50 mM HEPES-NaOH pH 7.5, 5 mM magnesium acetate, and 10 mM DTT and dialyzed overnight against 100 mM potassium acetate, 50 mM HEPES-NaOH pH 7.5, 5 mM magnesium acetate, and 10 mM DTT. Pol I was tested to be active in an RNA extension assay with this template (Tafur et al, 2016).

## Negative stain electron microscopy

The negatively stained samples were prepared on carbon coated grids (Quantifoil). Grids were glow discharged for 10 s, and then 3.5 μl of the sample (0.025 mg/ml) was deposited on the grids and incubated for 1 min. Grids were sequentially washed and stained with 1% (w/v) uranyl acetate solution and air-dried. Data of Pol I PIC (+DNA) were acquired with Tecnai Polara operating at 100 keV and magnification of 78,000 (1.9 Å pixel size), and data of Pol I PIC (−DNA) were acquired with Tecnai T12 operating at 120 keV and magnification of 68,000 (1.6 Å pixel size). The images were acquired in a defocus range of −1.5 to −2.0 μm and electron dose of 20 $e^-/Å^2$ using a 4k × 4k CCD Ultrascan camera.

## Cryo-electron microscopy

Cryo samples were prepared on holey copper grids (Quantifoil, R 2/1). The grids were glow discharged for 30 s using PELCO easyGlow. 2.5 μl of sample was applied on the grids, incubated for 15 s at 20°C in 100% humidity, and blotted for 8 s using a Vitrobot Mark II (FEI). Data were acquired on a Titan Krios (FEI) at 300 keV through a Gatan Quantum 967 LS energy filter using a 20 eV slit width in zero-loss mode. A total of 4,235 movie frames were recorded for the Pol I PIC on a Gatan K2-Summit direct electron detector at a nominal EFTEM (energy-filtered transmission electron microscope) magnification of 105,000×, corresponding to 1.35 Å calibrated pixel size (in 4K mode). The movies were collected within −0.8 to −4.0 μm defocus range in 20 frames with a dose rate of 2 $e^-/Å^2/s^1$ accumulating a total dose of 40 $e^-/Å^2$. Data collection was fully automated using SerialEM (Mastronarde, 2005).

## Data processing

The negative stain datasets were processed using EMAN2 and RELION-1.3 (Tang et al, 2007; Scheres, 2012). A total of 25,669 and 19,607 particles were semi-automatically picked using e2boxer.py from the EMAN2 package from Pol I PIC (+DNA) and Pol I PIC (−DNA), respectively. Particles were extracted with $240^2$ pixel box size and grouped by unsupervised 2D classification procedure using RELION. The selected classes representing the PIC were used to make an initial model with EMAN2 (e2initialmodel.py). The initial model was used as a reference for 3D classification and refinement in RELION, which resulted in a reconstruction of the Pol I PIC at 27 Å, which served as a starting model for the cryo-EM dataset.

For the cryo-EM dataset, the dose-fractionated movie frames were processed on-the-fly using UCSFImage4 to motion-correct and sum them (Li et al, 2015). The contrast transfer function (CTF) parameters of the micrographs were estimated using CTFFIND4 (Rohou & Grigorieff, 2015). The Thon rings were manually inspected, and micrographs with a poor fit were excluded from further analysis (154 micrographs). Approximately 30,000 particles were manually picked using e2boxer.py (EMAN2) and subjected to reference-free 2D classification in RELION-1.4. Classes that represented Pol I alone or PIC complex were chosen as templates for the autopicking procedure. A total of 867,673 particles were picked, extracted with $288^2$ box size, and sorted with 2D classification. A total of 508,049 particles from 2D classes with high-resolution features were refined using auto-refine against the initial model obtained from negative stain dataset. The same particles were also extracted with a smaller box size and used for processing the Pol I structures published recently (Tafur et al, 2016). Next, the alignment parameters were used to 3D classify particles, which resulted in one class with 46,071 PIC particles. These particles were subjected to 2D classification to remove the bad particles. In parallel, selected particles after 2D classification were subjected directly to 3D classification with alignment. This method resulted in one class with 42,131 PIC particles. Subsequently, unique particles selected from both methods were pooled, refined. A next round of

classification resulted in a major class (58,458 particles), which showed a better density for CF. Particles were further sorted using sequential rounds of auto-refine and 3D classification using masks on CF, the upper part of CF (the density above the upstream DNA), the lower part of CF, and Pol I-Rrn3. Finally, selected particles (38,589) were refined and post-processed to 4.4 Å. Refinement was also performed using either a mask on CF (CF-focused) or Pol I-Rrn3 (Pol I-focused) to improve the resolution of different parts of the map. The Pol I-focused refinement increased the resolution to 3.85 Å, enhancing the visibility of side chains and overall protein density for Pol I, but not for the single-stranded DNAs or Rrn3. Refinement focusing on CF improved the density for the upstream DNA and parts of the CF. However, flexible areas did not show better resolvability. The local resolution was estimated using Blocres, and the maps were filtered using Blocfilt (Cardone *et al*, 2013).

### Cross-linking and mass spectrometry

50 μg (1 mg/ml) of purified Pol I PIC was cross-linked by addition of an iso-stoichiometric mixture of H12/D12 isotope-coded, di-succinimidyl-suberate (DSS, Creative Molecules) to the final concentration of 2 and 5 mM. The cross-linking reactions were allowed to proceed for 30 min at 37°C and quenched by addition of ammonium bicarbonate to a final concentration of 50 mM for 10 min at 37°C. Cross-linked proteins were denatured using urea and RapiGest (Waters) at a final concentration of 4 M and 0.05% (w/v), respectively. Disulfide bonds were reduced using 10 mM DTT (30 min at 37°C), and cysteines were carbamidomethylated with 15 mM iodoacetamide (30 min in the dark). Protein digestion was performed first using 1:100 (w/w) LysC (Wako Chemicals GmbH, Neuss, Germany) for 3.5 h at 37°C then followed 1:50 (w/w) trypsin (Promega GmbH, Mannheim, Germany) overnight at 37°C, after the urea concentration was diluted to 1.5 M. Samples were then acidified with 10% (v/v) TFA and desalted using MicroSpin columns (Harvard Apparatus). Cross-linked peptides were enriched using size exclusion chromatography (Leitner *et al*, 2012). In brief, desalted peptides were reconstituted with SEC buffer (30% (v/v) ACN in 0.1% (v/v) TFA) and fractionated using a Superdex Peptide PC 3.2/30 column (GE) on an Agilent 1200 Infinity HPLC System at a flow rate of 50 ml/min. Fractions eluting between 1 and 1.8 ml were evaporated to dryness and reconstituted in 50 μl 5% (v/v) ACN in 0.1% (v/v) FA.

Fractions were analyzed by liquid chromatography (LC)-coupled tandem mass spectrometry (MS/MS) using a nanoAcquity UPLC system (Waters) connected online to LTQ-Orbitrap Velos Pro instrument (Thermo). Peptides were separated on a BEH300 C18 (75 × 250 mm, 1.7 mm) nanoAcquity UPLC column (Waters) using a stepwise 60- and 120-min gradient between 3 and 85% (v/v) ACN in 0.1% (v/v) FA. Data acquisition was performed using a TOP-20 strategy where survey-MS scans (m/z range 375–1,600) were acquired in the Orbitrap (R = 30,000) and up to 20 of the most abundant ions per full scan were fragmented by collision-induced dissociation (normalized collision energy = 40, activation Q = 0.250) and analyzed in the LTQ. In order to focus the acquisition on larger cross-linked peptides, charge states 1, 2, and unknown were rejected. Dynamic exclusion was enabled with repeat count = 1, exclusion duration = 60 s, list size = 500, and mass window ± 15 ppm. Ion target values were 1,000,000 (or 500 ms maximum fill time) for full

scans and 10,000 (or 50 ms maximum fill time) for MS/MS scans. To assign the fragment ion spectra, raw files were converted to centroid mzXML using the Mass Matrix file converter tool and then searched using xQuest (Leitner *et al*, 2014) against a fasta database containing the sequences of the cross-linked proteins. Posterior probabilities were calculated using xProphet (Walzthoeni *et al*, 2012).

### Modeling

The initial models of Rrn7 and the β-propeller domain of Rrn6 were built by homology modeling. The model of Rrn7 was built based on TFIIB (PDB ID 4v1n). The model can be deemed confident in the assignment of the fold and secondary structure. The exact sequence register, however, is uncertain for the first and the last helix of the N-terminal cyclin domain and the last two helices of the C-terminal cyclin domain, due to low sequence similarity of those regions to TFIIB. The N-terminal cyclin domain has a similar helical architecture as predicted before (Knutson *et al*, 2014). However, the C-terminal cyclin domain has an insertion of around five helices after the third helix, followed by two helices comprising residues 470–514, resembling a helix-turn-helix motif according to MODexplorer (Kosinski *et al*, 2013) and HHpred (Soding *et al*, 2005). The optimal modeling template for Rrn6 was identified by fitting all β-propellers from the CATH database to the EM map using the PowerFit software (van Zundert *et al*, 2016). The best fitting structure (β-propeller domain of Human Groucho/TLE1, PDB ID 1gxr, Fig EV4D) was used as a template for homology modeling, using alignments to various β-propeller proteins from MODexplorer (Kosinski *et al*, 2013). Although the sequence similarity to any β-propeller with known structure is low, the sequence alignments allowed for a tentative assignment of the WD40 motif for most of the repeats. All homology modeling was performed using MODexplorer (Kosinski *et al*, 2013), HHpred (Soding *et al*, 2005), GeneSilico Metaserver (Kurowski & Bujnicki, 2003), and Modeller (Webb & Sali, 2014).

Fitting of the cyclin domains of Rrn7 was performed using UCSF Chimera global search (Pettersen *et al*, 2004) and PowerFit, using Cα atoms as a query model and normalized local cross-correlation coefficient as a fitting measure. Both programs led to equivalent fits. All helices of the cyclin domains were identifiable in the density. Slight adjustments based on the EM density where performed manually using Coot (Emsley & Cowtan, 2004) to optimally fit the helices and connecting loops. The Zn-ribbon was placed using the position of the TFIIB B-ribbon in the Pol II PIC (PDB ID 5fyw) as a template, by alignment on the second largest subunit of Pol I (A135) and II (Rpb2). The orientation of the homology model for the Rrn7 Zn-ribbon was manually adjusted in Coot (Emsley & Cowtan, 2004). The Rrn6 β-propeller domain has been placed according to the fit to the density, but its exact orientation is uncertain.

The remaining regions of Rrn7 and some regions of Rrn11 were then built manually using Coot (Emsley & Cowtan, 2004) wherever the assignment was reliable. Two helical densities were tentatively assigned to the insertion in the C-terminal cyclin domain (residues 318–472) of Rrn7 that cross-links to Rpb10 and Rpb12. Nine helices comprising residues 204–442 could be assigned to Rrn11. The tracing of the Rrn11 TPR domain was additionally supported by fits of unrelated TPR domains derived from CATH (Orengo *et al*, 1997) and SCOP database (Murzin *et al*, 1995), automatically

fitted to the EM map using PowerFit (van Zundert *et al*, 2016). In addition, the cross-links from Lys333 of Rrn11 to Lys30 of A135 and Lys406 of Rrn11 to Lys174 of A135 supported the placement of the TPR domain. Altogether, most of the CF density has been assigned to the corresponding subunits except the N-terminal helices of Rrn11 (residues 1–203) and the helical domain of Rrn6. There are several helical densities in our EM density around cyclin domains of Rrn7, and the DBHB positioned between the β-propeller and the upstream DNA that we could not assign. The helical densities located around the cyclin domains should correspond to the helical domain of Rrn6 according to previously published cross-links (Knutson *et al*, 2014) and cross-links identified in this work (Fig EV4A and B).

The upstream DNA was modeled by generating an ideal B-DNA template using the 3D-DART server (van Dijk & Bonvin, 2009) and adjusting it manually based on the density using Coot. Base pair numbering was tentatively assigned by placing the base pair −50 at the upstream end of the DNA density observable in the map. Since the density of approximately the first five base pairs at the upstream end was weakly resolved, only the DNA from the base pair −45 was included in the final model. The downstream DNA from the Pol I OC (PDB ID 5m5w) was rigid body fitted into the density and manually adjusted in Coot. Figures were prepared using UCSF Chimera (Pettersen *et al*, 2004). The electrostatic potential distribution was calculated using APBS (Baker *et al*, 2001) and visualized using the PyMOL Molecular Graphics System (Version 1.8 Schrödinger, LLC.).

Rrn3 was first fitted using the orientation relative to Pol I based on the Pol I-Rrn3 complex (PDB ID: 5g5l) and was manually adjusted as a rigid body to fit the density in Coot. Due to the low resolution of this region, no further modifications in Rrn3 were made.

For Pol I, the apo crystal structure was divided into previously described modules (Fernandez-Tornero *et al*, 2013; Tafur *et al*, 2016) and rigid body fitted into the density for the Pol I-focused map in UCSF Chimera. Individual regions were manually adjusted while regions not visible in the density were deleted in Coot.

### Accession codes

The cryo-EM densities for the Pol I PIC, Pol I PIC (CF-upstream DNA focused refinement), and Pol I PIC (Pol I-Rrn3 focused refinement) have been deposited in the EMDB with accession codes EMD-3727, EMD-3728, and EMD-3729, respectively. The coordinates of the Pol I PIC have been deposited in the PDB with accession code 5OA1. The mass spectrometry proteomics data have been deposited to the ProteomeXchange Consortium via the PRIDE partner repository (Vizcaíno *et al*, 2016) with the dataset identifier PXD006510.

Expanded View for this article is available online.

### Acknowledgements

We thank N. A. Hoffmann and P. Kastritis for advice in data processing, S. Mosalaganti for help with cryo-EM data acquisition, and B. Murciano for help in sample preparation. Y.S, L.T, R.W, and C.W.M acknowledge support by the ERC Advanced Grant (ERC-2013-AdG340964-POL1PIC). K.B acknowledges support by the EMBL International PhD Program. J.K and A.J.J acknowledge support by the EMBL Interdisciplinary Postdoc Program (EIPOD) under Marie Curie COFUND actions (PCOFUND-GA-2008-229597). A.J.J acknowledges financial support by the Deutsche Forschungsgemeinschaft (DFG) through the excellence cluster "The Hamburg Center for Ultrafast Imaging (CUI) – Structure, Dynamics and Control of Matter at the Atomic Scale" (EXC1074) and the Joachim Herz Foundation.

### Author contributions

CWM initiated and supervised the project. YS and LT designed and carried out experiments and data processing. YS and WJHH collected negative stain and cryo-EM datasets. YS, LT, AJJ, and CS analyzed the cryo-EM data. YS, LT, and JK carried out the modeling. KB performed and MB supervised the cross-linking experiment. RW carried out yeast fermentation and assisted in sample preparation. YS, LT, JK, and CWM prepared the manuscript with input from all other authors.

### Conflict of interest

The authors declare that they have no conflict of interest.

### Note added in proof

An additional cryo-EM structure of a *Saccharomyces cerevisiae* initial transcribing Pol I-Core Factor-DNA complex has recently been determined at 3.8 Å resolution (Han *et al*, 2017). The overall architecture of this complex is very similar to our Pol I pre-initiation complex and the Pol I initial transcribing complex reported by Engel *et al* (2017), despite the fact that this complex does not contain the bridging factor Rrn3.

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
