## [Review Process File · The EMBO Journal]

Manuscript EMBO-2017-96958

Structural insights into transcription initiation by yeast RNA polymerase I

Yashar Sadian, Lucas Tafur, Jan Kosinski, Arjen J. Jakobi, Rene Wetzel, Katarzyna Buczak, Wim J. H. Hagen, Martin Beck, Carsten Sachse & Christoph W. Müller

Corresponding author: Christoph W. Müller, European Molecular Biology Laboratory

Review timeline:	Submission date:	17 March 2017
	Editorial Decision:	05 April 2017
	Revision received:	26 May 2017
	Editorial Decision:	16 June 2017
	Revision received:	20 June 2017
	Accepted:	26 June 2017

Editor: Anne Nielsen

Transaction Report:

1st Editorial Decision

05 April 2017

Thank you for submitting your manuscript for consideration by the EMBO Journal. It has now been seen by an expert referee and this person's comments are shown below. Given the competitive situation and since our referee is supportive of publication following textual revision only, we have decided not to involve additional referees.

As you will see from the report, the referee highlights the quality and relevance of your data and supports publication after adequate revision. The referee also mentions the study on initiating pol I from Engels et al that appeared after your manuscript was submitted here - and asks you to take the conclusions from that study into account when preparing a revised manuscript.

Given the referee's positive recommendation, I would like to invite you to submit a revised version of the manuscript, in which you follow the second option proposed by the referee and revise the discussion section to acknowledge the findings from Engels et al. You will also have to address the various major and minor concerns raised by the referee, mainly via additional discussion and clarification. I should add that it is EMBO Journal policy to allow only a single round of revision, and acceptance of your manuscript will therefore depend on the completeness of your responses in this revised version.

Thank you for the opportunity to consider your work for publication. I look forward to your revision.

REFEREE REPORT

Referee #1:

Sadian et. al present the cryo-EM reconstruction of a "pre-initiation complex" (PIC) of RNA polymerase I with bound Rrn3 and core factor. Particles were extensively classified to obtain an electron density of high quality that enabled unambiguous placement of known structures of Pol I and Rrn3 and assignment of secondary structure elements to the core factor density. The authors discuss differences to the Pol II and Pol III transcription systems. A hypothetical mechanism of transition between a Pol II-like and a Pol I specific PIC is presented but lacks evidence. The results are of strong interest to the transcription community. The EM data are of high quality but could not be fully explored because details of the core factor interactions with Pol II and DNA could not be revealed due to a lack of a high-resolution core factor structure.

Only two days after the manuscript was available to this reviewer, a similar study was published (Engel et al., Cell 2017) that presents the core factor crystal structure and the cryo-EM structure of the Pol I initially transcribing complex (ITC) with bound Rrn3 and core factor. This paper must now be taken into account. This reviewer can see two possible ways of doing so. First, the authors could include the new paper in the introduction, use the core factor crystal structure for the interpretation of their cryo-EM reconstruction, and rewrite the results and discussion to fully explore the implications of their EM work by making use of the results in Engel et al., in particular the core factor crystal structure. In this case, the resulting manuscript would have to be treated as a new submission, with a new submission date, because several conclusions would likely change and new paragraphs would likely be added that relied on the Engel et al. paper. Alternatively, the authors could revise the current manuscript. In this case, the Engel et al. paper should be mentioned only in the discussion ('After our work had been submitted, a study was published...'). The authors would then have to add text to the discussion that would describe the similarities and differences to the recently published ITC structure. Most importantly, the authors would have to mention whether the overall architecture of their PIC and the ITC in Engel et al. similar and the core factor is positioned similarly on Pol I. To preserve independency, the latter option will not allow the authors to edit their results section based on insights and results from the Engel et al. paper. A good solution could be to start the discussion with a summary of the findings and a statement that the Engel et al. paper came out, followed by an interpretation taking the newly published work into account, and concluding with similarities, differences, and open questions for the future.

Taken together, this reviewer strongly recommends publication because the presented results were derived independently and are important. Publication in The EMBO Journal however depends on a positive re-evaluation of the manuscript after an appropriate way of including the results from Engel et al. has been found and after the below concerns have been addressed.

Major concerns

1. Identity of the complex studied by EM.

The authors state that they present the 'molecular architecture of the yeast Pol I pre-initiation complex'. However, they actually subjected to analysis an initially transcribing complex with DNA and 10 nt RNA, which is by definition post initiation of the RNA chain. The RNA may have been lost or degraded by the intrinsic Pol I cleavage activity. Alternatively, the DNA-RNA hybrid may be present but invisible, maybe because it is not firmly positioned due to incomplete compaction of the active center cleft (compare concern 2 below). Are the authors certain that the final reconstruction represents an OC, provided a large mismatch bubble is present? Is it possible that the 10 nt RNA interferes with binding of the Rrn7 zinc ribbon to the Pol I dock domain and the neighboring Rrn3, leading to weaker density for the zinc ribbon and Rrn3? The authors must describe this uncertainty with respect to the RNA and with respect to complex identity and should consequently reflect whether the title of the manuscript is accurate.

2. Apparent cleft closure during the OC-ITC transition.

Related to the first concern, the current interpretation of the authors 'PIC' density suggests that the OC has a rather open/expanded cleft, whereas we know the ITC and EC have more closed/contracted clefts. Figure EV7 compares cleft closure between the Pol I OC, EC and PIC. Whereas cleft closure is obvious in the EC, the OC and PIC adopt more open conformations. This, if confirmed, is an interesting new point that the authors should discuss. However, how do the authors envision RNA synthesis can commence in an open cleft when the template DNA strand must be positioned in the active center such that it allows for stable positioning of NTP substrates at the active site magnesium ion to enable catalysis and binding of short RNA chains in the initially transcribing complex? In Figure 2C the single-stranded template DNA in the PIC seems to be tilted with respect to the template DNA in the EC that contains a DNA-RNA hybrid. Do the authors argue that binding of the DNA-RNA hybrid induces cleft closure? Related to this, the authors should comment on the occupancy and orientation of the A12.2 C-terminal domain. Is it present at all? Does it show low occupancy, similar to the Rrn7 zinc ribbon? Note that presence of the A12.2 C-terminal domain in the pore of the polymerase does not allow for complete cleft contraction and therefore the active center is very likely not in a conformation that allows for catalysis. Taken together, if the reconstruction represents an OC, the more open conformation is unexpected and it is not obvious how it would support RNA synthesis to lead to an ITC. This point may be addressed in the discussion when comparing results presented here to the ITC structure presented by Engel et al.

3. Role of TBP in Pol I initiation and the hypothesis of an 'early PIC'.

The role of TBP in the Pol I initiation system remains to be clarified. The authors speculate about the role of TBP in the results section without presenting data. They also hypothesize that there is an early PIC that substantially deviates from the structure they observe here, and present a supplemental movie to illustrate the major changes that one would expect from a conversion of such a hypothetical early PIC to the observed structure. This modeling is not justified by data; and to this reviewer it looks as if TBP binding to the position indicated in Fig. 4 would lead to major clashes due to DNA bending and direct CF clashes. To justify such modeling, the authors must at least show that TBP can be incorporated stably into the PIC without UAF and that it is positioned where they expect it to be using crosslinking and mass spectrometry. If such data or similar data are not available, the speculation should be removed from results and the supplemental movie should not be part of the published manuscript, to avoid confusion. The authors may of course reflect on the possibility of TBP bending the DNA in the Pol I system in the discussion. However, if the authors consider this, they should state whether TBP alone can bind to the presented complex. This reviewer doubts this because TBP alone does not influence initiation *in vitro*. It is however known that TBP influences initiation in the context of UAF (Keener et al. 1998). Including these points in a discussion paragraph would be a good way of dealing with the role of TBP in Pol I initiation. Thus, the entire discussion paragraph on the early PIC should be rewritten taking these considerations into account.

Minor concerns

4. abstract - 'ribosomal genes are transcribed by RNA polymerase I' - this is not really correct (consider ribosomal protein genes, 5S rRNA)
5. Page 4: '...,in a conformation different from previously observed bacterial or eukaryotic Pol II PICs.' It might help the reader to better understand the comparison between bacterial PIC and Pol I PIC, if the bacterial PIC was introduced before.
6. Can the authors explain why CF did not stably bind to Pol I in the presence of Rrn3 without a DNA scaffold under their experimental conditions?
7. The authors use the PDB 5G5L (which has a lower resolution than the presented reconstruction) to position Rrn3. Why did they chose this PDB file and is there a difference in Rrn3 orientation in their density? This should be mentioned.
8. Page 7: 'The upstream DNA also comes from another loop that...' Please, specify the loop.
9. If the density for the Rrn7 zinc ribbon is largely missing (Fig. 1), modelling of this domain seems arbitrary and should not be done to interpret this reconstruction.
10. What do the authors mean by 'CF does not induce any major conformational changes in Pol I'? Is that in comparison to the Pol I-Rrn3 complex? Observations in Fig. EV7C should be mentioned in the main text.
11. Page 8/9 - 'The conserved position of the Zn-ribbon next to the RNA exit channel suggest that despite the topological differences with Pol II, Rrn7 might function in a similar manner to TFIIB...'

Could it also be the case that the conserved Zn ribbons of TFIIB and Rrn7 function in a similar manner, while the cyclin domains function in a different way in the Pol I system than they do in the Pol II system?

12. Transcriptional activity: Adding a sentence that the proteins were previously tested for activity (Tafur et al., 2016) would increase the relevance.
13. Speculations concerning Pol I adaptation and promoter escape must be moved from the results to the discussion.
14. From the figures it does not appear that the DBHB really binds the major groove of DNA in a way expected from major groove binding factors. In fact, a helix rather seems to contact the DNA backbone. This should be rephrased.
15. Which positive patches in the Rrn11 TPR domain 'form a continuous positively charged binding surface along the DNA path'? How do the authors know which residues are on the surface?
16. Details on crosslinking (number of crosslinks, cut-off criteria, sample preparation, etc.) should be added to the main manuscript if they are being presented in the supplement.
17. Why are there differences in the cross-link presentation in EV4 A and B (eg. Rrn11 K406 crosslinks in panel B but not in panel A).
18. Are there crosslinks for the tandem winged helix domain of A49? Furthermore, is there any density for this domain in a cryo-EM class that agrees with one of the two presented positions (Pils et al., 2016 and Tarfur et al, 2016)?
19. PDB/EMDP submission - The authors should make their models available for the public.
20. Figures would be more easily accessible if the views were always related.
21. Figure 2 - In panel A there seems to be a gap between protrusion- and DBHB-DNA contacts. Yet, panel B shows overlapping DNA contacts. Can this be explained?
22. Figure 2B - The rDNA core promoter spans from -50 to +5, but appears it is -38 to +5 (Keener et al. 1998)?
23. Figure 5 - This figure may be improved. First, A12.2 and Rrn11 are both depicted in yellow. Second, stalk and A49-A34.5 may be labeled for orientation. Third, while cleft closure is discussed in the text, it is omitted from the figure. Finally, RNA is included in the bottom panel of the EC, but the cartoon representation does not show RNA.
24. "Unusual DNA position in the Pol I PIC" - unusual with respect to what? Maybe better say 'Unexpected'.
25. Page 6. What do the authors mean by fold recognition. This should be explained.
26. What is the A135 'wedge'? If it is a loop, it should be named according to flanking secondary structure elements so that it can be easily identified.
27. First sentence of discussion: it is unclear what 'first' refers to, the PIC or the open state. This should be rephrased, now taking into account the Engel et al. paper.

1st Revision - authors' response

26 May 2017

Point-by-Point-Response

We would like to thank the reviewer for the critical reading of our manuscript and the insightful comments. We have addressed the reviewer's points as detailed below.

Major concerns of the referee

1. Identity of the complex studied by EM.

The authors state that they present the 'molecular architecture of the yeast Pol I pre-initiation complex'. However, they actually subjected to analysis an initially transcribing complex with DNA and 10 nt RNA, which is by definition post initiation of the RNA chain. The RNA may have been lost or degraded by the intrinsic Pol I cleavage activity. Alternatively, the DNA-RNA hybrid may be present but invisible, maybe because it is not firmly positioned due to incomplete compaction of the active center cleft (compare concern 2 below). Are the authors certain that the final reconstruction represents an OC, provided a large mismatch bubble is present? Is it possible that the 10 nt RNA interferes with binding of the Rrn7 zinc ribbon to the Pol I dock domain and the neighboring Rrn3, leading to weaker density for the zinc ribbon and Rrn3? The authors must describe this uncertainty with respect to the RNA and with respect to complex identity and should consequently reflect whether the title of the manuscript is accurate.

Response: We thank the referee for these insightful comments. Indeed, we setup our experiment to obtain an initially transcribing Pol I complex. However, for unknown reasons our final reconstruction contains a DNA transcription scaffold, but no density for RNA indicating that RNA has been lost or degraded. Concerning the identity of the complex, we interpret our Pol I complex as an OC with open cleft, partially unfolded BH and A12.2 C-terminal domain in the active site that still has to transition into a Pol I ITC as reported by Engel et al. 2017. In support of this description, we do not think that the size of the mismatch in our bubble interferes with the formation of an OC, because a similar bubble with 15-nt mismatch has been used to capture the OC in yeast Pol II (Plaschka et al, 2016) and with a 13-nt mismatch bubble to capture the human Pol II OC (He et al., 2016). Although we favor the description of our complex as a Pol I OC, alternatively, the observed complex could also represent a “post-cleaved” state in which the Pol I A12.2 C-terminal domain has cleaved the RNA (reminiscent of the human Pol II initiation complex in the presence of TFIIS (He et al. 2016), where also no density for RNA was observed). The third suggestion of the referee that the 10-nt RNA interferes with the binding of the Rrn7 Zn-ribbon to the Pol I dock domain and destabilizes Rrn3 is probably less likely. Mainly, because i) Pol I is in an pre-transcribing rather than a transcribing conformation, ii) because we see clear density for the zinc ribbon domain at the Pol I dock domain (Figure 3, Figure for reviewer, B) and iii) because we explain the smearing out of density of Rrn3 rather by a rotational movement of Rrn3 relative to Pol I rather than complete dissociation.

At present, we cannot really distinguish between an “OC” or “post-cleaved” state of our Pol I complex. In fact, we think that our complex represents an intermediate situation that can either transition towards active transcription or towards abortion of transcription. Therefore, both states might be structurally indistinguishable. Accordingly, we discuss both possibilities in the revised version of the manuscript on page 12/13. Despite the uncertainty about the complex identity, we continue to refer to the complex as “pre-initiating complex, PIC” to distinguish it from the “initially transcribing complex, ITC” described by the Cramer lab, but we also discuss the difficulty exactly defining the complex identity on page 12/13. Because of this difficulty, we also followed the suggestion of the referee and changed the title of our manuscript into “Structural insights into transcription initiation by yeast RNA polymerase I”.

2. Apparent cleft closure during the OC-ITC transition.

Related to the first concern, the current interpretation of the authors 'PIC' density suggests that the OC has a rather open/expanded cleft, whereas we know the ITC and EC have more closed/contracted clefts. Figure EV7 compares cleft closure between the Pol I OC, EC and PIC. Whereas cleft closure is obvious in the EC, the OC and PIC adopt more open conformations. This, if confirmed, is an interesting new point that the authors should discuss. However, how do the authors envision RNA synthesis can commence in an open cleft when the template DNA strand must be positioned in the active center such that it allows for stable positioning of NTP substrates at the active site magnesium ion to enable catalysis and binding of short RNA chains in the initially transcribing complex? In Figure 2C the single-stranded template DNA in the PIC seems to be tilted with respect to the template DNA in the EC that contains a DNA-RNA hybrid. Do the authors argue that binding of the DNA-RNA hybrid induces cleft closure?

Response: We have now modified and expanded the discussion to discuss these issues in greater detail (page 13). We observe indeed that in our Pol I PIC (and as well as in the Pol I OC in the absence of CF and Rrn3 (Tafur et al, 2016) the DNA-binding cleft is wider and the bridge helix unwound compared to the Pol I ITC (Engel et al. 2017) and the Pol I EC (Tafur et al, 2016, Neyer et al, 2016). In addition, the position of the template DNA strand is also different from the Pol I ITC and Pol I EC (compare Figure for reviewer, A). This suggests that our Pol I complex would still have to undergo some conformational changes to transition to the Pol I ITC unlike as in the Pol II system, where OC, ITC and EC adopt very similar conformations. The required changes in Pol I (cleft closing, bridge helix folding, displacement of A12.2C from the active site, and repositioning of the single-stranded template DNA strand) could be indeed promoted by the stable binding of a DNA-RNA hybrid as suggested by the referee, although other factors might also contribute to this transition.

Referee: Related to this, the authors should comment on the occupancy and orientation of the A12.2 C-terminal domain. Is it present at all? Does it show low occupancy, similar to the Rrn7 zinc ribbon? Note that presence of the A12.2 C-terminal domain in the pore of the polymerase does not allow for complete cleft contraction and therefore the active center is very likely not in a

conformation that allows for catalysis. Taken together, if the reconstruction represents an OC, the more open conformation is unexpected and it is not obvious how it would support RNA synthesis to lead to an ITC. This point may be addressed in the discussion when comparing results presented here to the ITC structure presented by Engel et al.

Response: In the final reconstruction, the A12.2 C-terminal domain is present in the active site although the corresponding density is slightly weaker than for neighboring regions suggesting increased flexibility. We have now included the density at a lower threshold in Fig. EV5A. Certainly, the presence of the A12.2 C-terminal domain prevents cleft closure and is reminiscent of the situation in the Pol I OC observed by Tafur et al, 2016. As discussed above, the changes required for formation of the ITC could be promoted by stable binding of a DNA-RNA hybrid although other factors might also contribute. A different Pol I OC conformation with a more expanded cleft compared to the Pol I ITC and the Pol I EC also agrees with the observation that in Pol I, the middle region of the bridge helix can unwind while this has not been observed in Pol II. We have also added this point to the discussion.

3. Role of TBP in Pol I initiation and the hypothesis of an 'early PIC'.

The role of TBP in the Pol I initiation system remains to be clarified. The authors speculate about the role of TBP in the results section without presenting data. They also hypothesize that there is an early PIC that substantially deviates from the structure they observe here, and present a supplemental movie to illustrate the major changes that one would expect from a conversion of such a hypothetical early PIC to the observed structure. This modeling is not justified by data; and to this reviewer it looks as if TBP binding to the position indicated in Fig. 4 would lead to major clashes due to DNA bending and direct CF clashes. To justify such modeling, the authors must at least show that TBP can be incorporated stably into the PIC without UAF and that it is positioned where they expect it to be using crosslinking and mass spectrometry. If such data or similar data are not available, the speculation should be removed from results and the supplemental movie should not be part of the published manuscript, to avoid confusion. The authors may of course reflect on the possibility of TBP bending the DNA in the Pol I system in the discussion. However, if the authors consider this, they should state whether TBP alone can bind to the presented complex. This reviewer doubts this because TBP alone does not influence initiation *in vitro*. It is however known that TBP influences initiation in the context of UAF (Keener et al. 1998). Including these points in a discussion paragraph would be a good way of dealing with the role of TBP in Pol I initiation. Thus, the entire discussion paragraph on the early PIC should be rewritten taking these considerations into account.

Response: In the revised version of the manuscript we have followed the suggestion of the referee and removed the paragraph about TBP binding and the hypothetical "early" PIC because it is indeed speculative and not supported by experimental data. Instead, we have focused the discussion on the identity of our Pol I initiation complex and compared it to the Pol ITC. Accordingly, we modified Fig. 5 and removed the supplemental movie.

Minor concerns

4. abstract - 'ribosomal genes are transcribed by RNA polymerase I' - this is not really correct (consider ribosomal protein genes, 5S rRNA)

Response: We have corrected the abstract that now states more precisely: "*In eukaryotic cells, RNA polymerase I (Pol I) synthesizes precursor ribosomal RNA (pre-rRNA) that is subsequently processed into mature rRNA.*"

5. Page 4: '...,in a conformation different from previously observed bacterial or eukaryotic Pol II PICs.' It might help the reader to better understand the comparison between bacterial PIC and Pol I PIC, if the bacterial PIC was introduced before.

Response: We have now introduced the bacterial PIC as follows: "*While Pol II requires the assembly of several general transcription factors for transcription initiation, bacterial Pol only requires the action of sigma factor (Feklistov et al, 2014)*"

6. Can the authors explain why CF did not stably bind to Pol I in the presence of Rrn3 without a DNA scaffold under their experimental conditions?

Response: We observe that the Pol I PIC did not form in the absence of DNA, which is in contrast to the results reported by Engel et al, 2017. We speculate that this difference results from the different experimental conditions used during complex formation. We used an equimolar ratio of CF to Pol I and a small excess of Rrn3 (2x) in 100 mM KAc without crosslinking our sample, while Engel et al. incubated Pol I with a bigger excess of Rrn3 and CF (5x) followed by gel filtration and crosslinking. This suggests that in the absence of crosslinking and without an excess of CF relative to Pol I, DNA is required to form a stable Pol I complex.

7. The authors use the PDB 5G5L (which has a lower resolution than the presented reconstruction) to position Rrn3. Why did they chose this PDB file and is there a difference in Rrn3 orientation in their density? This should be mentioned.

Response: We chose this PDB entry as we tried to keep Rrn3 in the same position relative to Pol I as much as possible. Given that the Rrn3 density is of lower quality than the density for Pol I, this approach appeared to us more conservative and more accurate than independently fitting Rrn3. Compared to the Rrn3 position in the Pol I ITC (Engel et al, 2017), the position of Rrn3 relative to Pol I is very similar in our Pol I PIC. However, we noticed that Rrn3 is slightly rotated with respect to Pol I as illustrated in Figure EV5F. The fact that we used PDB 5G5L is now mentioned in the Materials and Methods section, page 23.

8. Page 7: 'The upstream DNA also comes from another loop that...' Please, specify the loop.

Response: The loop has now been specified on page 7: *The upstream DNA also comes closer to another loop of A135 (residues 890-897) that extends from the A135 wall that probably stabilizes the DNA position (Fig EV5B).*

9. If the density for the Rrn7 zinc ribbon is largely missing (Fig. 1), modelling of this domain seems arbitrary and should not be done to interpret this reconstruction.

Response: The density for the Rrn7 zinc ribbon is missing in Fig. 1 only due to the high threshold used in this figure to better visualize secondary structure elements. It is clearly visible at a higher threshold as shown in Fig. 3A & B. The density allowed unambiguous fitting of the Rrn7 zinc ribbon that matches with the position and orientation of the TFIIB zinc ribbon and the Rrn7 zinc ribbon modelled in the ITC (Engel et al, 2017). To explain this, we have added a sentence in the figure legend of Fig.1: *“The threshold of the cryo-EM map was chosen to clearly visualize secondary structure elements. The density for the Rrn7 Zn ribbon is not visible at this threshold but is depicted in Fig. 3A and B.”*

10. What do the authors mean by 'CF does not induce any major conformational changes in Pol I'? Is that in comparison to the Pol I-Rrn3 complex? Observations in Fig. EV7C should be mentioned in the main text.

Response: We referred to the comparison with the Pol I OC in the absence of CF and Rrn3 (Tafur et al, 2016). To clarify this point we have changed the text on page 11 that now reads: *“Apart from the movement of this loop, Pol I retains essentially the same conformation as seen in the Pol I open complex (OC) (PDB ID: 5m5w, Tafur et al, 2016), where Pol I has an open cleft, contains the C-terminus of A12.2 in the active site, and where the bridge helix is not fully folded (Fig EV5C).”*

11. Page 8/9 - 'The conserved position of the Zn-ribbon next to the RNA exit channel suggest that despite the topological differences with Pol II, Rrn7 might function in a similar manner to TFIIB...' Could it also be the case that the conserved Zn ribbons of TFIIB and Rrn7 function in a similar manner, while the cyclin domains function in a different way in the Pol I system than they do in the Pol II system?

Response: This is indeed possible and we have added this point to the manuscript, but also moved it from the results to the discussion section, page 11/12: *“Moreover, the cyclin domains of Rrn7 are positioned differently than TFIIB, while the Zn-ribbon is present in a conserved position next to the RNA exit channel. This suggests that the N-terminal region of Rrn7 might function in a similar manner to TFIIB, as proposed previously (Knutson & Hahn, 2013), but that the cyclin domains could have different or additional functions compared to the Pol II system although they use similar interfaces as TFIIB and Brf2 to interact with promoter DNA”*

12. Transcriptional activity: Adding a sentence that the proteins were previously tested for activity (Tafur et al., 2016) would increase the relevance.

Response: We now include an extra sentence in the materials and method section, page 16: “*Pol I was tested to be active in an RNA extension assay with this template (Tafur et al, 2016).*”

And in the main text, page 5:

“We stepwise assembled the minimal Pol I PIC by adding recombinant CF, Rrn3 and natively purified, transcriptional active Pol I to a 70 base pair (bp) transcription scaffold (-50 to +20) containing the core rDNA promoter (-38 to +5) with a 15-nucleotide (nt) mismatch and a 10-nt RNA (Tafur et al, 2016).”

13. Speculations concerning Pol I adaptation and promoter escape must be moved from the results to the discussion.

Response: We agree with the referee and now moved this paragraph to the discussion (page 11).

14. From the figures it does not appear that the DBHB really binds the major groove of DNA in a way expected from major groove binding factors. In fact, a helix rather seems to contact the DNA backbone. This should be rephrased.

Response: We agree with the reviewer and we changed the text on page 9, which now reads:

“The DBHB contacts the DNA backbone opposite to the minor groove that is close to the Rrn7 cyclin domains.”

15. Which positive patches in the Rrn11 TPR domain 'form a continuous positively charged binding surface along the DNA path'? How do the authors know which residues are on the surface?

Response: In our model residues Arg206, Arg209, Arg241, Lys248, Lys281, Arg283, Arg291, Arg302, His304, and Lys307 contribute to the positive patch on Rrn11. We approximately assigned the sequence register of Rrn11 and could predict that these residues are located on the DNA binding surface of our model. Although the exact sequence register in the Rrn11 TPR domain in our model is uncertain, we are confident about the approximate sequence assignment of the helical densities of Rrn11 based on modeling in Coot, sequence connectivity, comparisons to TPR folds, and crosslinks (explained in the Methods). We cannot exclude minor sequence shifts but they would not affect the localization of these residues to the surface proximal to the DNA and the calculation of an approximate surface potential. Indeed, comparison of our model to the recently released crystal structure of CF by Engel et al, 2017 shows that these residues are placed in similar positions on the surface next to DNA (with only small differences resulting from minor shifts in the sequence register). We have now explained this in the revised manuscript more clearly on page 9: *“Although the exact sequence register in the Rrn11 TPR domain in our model is uncertain, the sequence for the helical densities in the TPR domain could be approximately assigned (see Material and Methods) and we only expect minor sequence register shifts. This allows calculating an approximate electrostatic potential, which is not significantly affected by minor sequence register shifts.”*

16. Details on crosslinking (number of crosslinks, cut-off criteria, sample preparation, etc.) should be added to the main manuscript if they are being presented in the supplement.

Response: We have moved this information from the Methods to the main text on page 6:

“To aid in modeling the remaining densities, we performed cross-linking mass spectrometry of purified Pol I PIC using the di-succinimidyl-suberate (DSS) crosslinker. We obtained 124 unique inter-links and 194 unique intra-links with the LD (linear-discriminant) confidence score, at least 23 (Fig EV4A), as calculated by xQuest (Leitner et al, 2014). The appropriateness of the score threshold was validated using the Pol I crystal structure (PDB ID 4c3i), for which 113 out of 116 crosslinks (97.4%) mapped to the structure satisfied the distance threshold of 30 Å. With the aid of the crosslinking data (Fig EV4B) and identification of macromolecular folds guided by the cryo-EM density (Fig EV4C and Methods), we could assign most of the Rrn11 helices, which resulted in a topological model of the Pol I PIC (Fig 1A).”

17. Why are there differences in the cross-link presentation in EV4 A and B (eg. Rrn11 K406 crosslinks in panel B but not in panel A).

Response: All crosslinks from panel B were present in the panel A, but some were not clearly visible due to an unfortunate choice of the yellow color for crosslinks originating from Rrn11. We have fixed it by changing the color for the crosslinks to black in Fig EV4A.

18. Are there crosslinks for the tandem winged helix domain of A49? Furthermore, is there any density for this domain in a cryo-EM class that agrees with one of the two presented positions (Pils et al., 2016 and Tarfur et al, 2016)?

Response: Yes, we observe six crosslinks from the tandem winged helix (tWH) domain of A49 to other subunits and several intra-A49 crosslinks. Three crosslinks (A49,410-A190,211, A49,K306-A190,267, and A49,K227-A190,1334) link to regions around the DNA binding cleft, one (A49,211-A43,K276) to the stalk, and two (A49,K223-Rrn3,22 and A49,211-Rrn3,22) to the N-terminus of Rrn3. In the revised version of the manuscript, we have uploaded a list of all crosslinks as Source Data 1.

We observe weak density for the tWH domain in a similar position as observed in the Pol I ITC by Engel et al., 2017, but we could not assign it unambiguously to the tWH (see also Figure for reviewer, panel C). We speculate that the density for the tWH domain is weak because of the open clamp position. In both, the EC_tWH (Tafur et al, 2016) and the ITC_tWH (Engel et al, 2017), the linker crossing the cleft is clearly visible. Because we observe a more open clamp conformation and no clear linker density in the cleft, we conclude that in the Pol I PIC the A49 tWH is more mobile. This is also consistent with the observation that the tWH domain was crosslinked to two regions distant from each other (the stalk and the cleft).

19. PDB/EMDP submission - The authors should make their models available for the public.

Response: We have now deposited the Pol I PIC, the Pol I PIC (CF-upstream DNA focused) and the Pol I PIC (Pol I-Rrn3 focused) cryo-EM maps in the EMD under accession codes EMD-3727, EMD-3728 and EMD-3729, respectively. Our strategy but also limitations in obtaining a CF model based on the 4.4 Å cryo-EM map are described in detail in the Material and Methods section. Because of the limitations of our current model (partially complete, minor sequence register shifts etc.) we prefer not to deposit the atomic coordinates of the Pol I PIC in the PDB as they could confuse rather than support users. Instead, the atomic coordinates of our Pol I PIC will be available upon request by the corresponding author.

20. Figures would be more easily accessible if the views were always related.

Response: The views in Fig. 1 are related to common views published earlier. However, to aid in the interpretation, we have included additional labels relating the Pol I views to the figures.

21. Figure 2 - In panel A there seems to be a gap between protrusion- and DBHB-DNA contacts. Yet, panel B shows overlapping DNA contacts. Can this be explained?

Response: There is indeed a gap between protrusion- and DBHB-DNA contacts, but in this region the DNA is contacted by other regions of Rrn11 on the opposite side of the DNA. Therefore, the entire DNA in this region is continuously contacted by either CF, Pol I or both. Because we are not sure about the exact sequence register of the DNA we indicated the uncertainty in the boundaries by dashed lines. This is now also clarified in the legend of Fig. 2: *“Dashed lines for CF and Pol I indicate uncertainty in the boundaries due to the unknown exact sequence register of the DNA”*

22. Figure 2B - The rDNA core promoter spans from -50 to +5, but appears it is -38 to +5 (Keener et al. 1998)?

Response: We apologize for this mistake and corrected Fig.2 accordingly.

23. Figure 5 - This figure may be improved. First, A12.2 and Rrn11 are both depicted in yellow. Second, stalk and A49-A34.5 may be labeled for orientation. Third, while cleft closure is discussed in the text, it is omitted from the figure. Finally, RNA is included in the bottom panel of the EC, but the cartoon representation does not show RNA.

Response: The figure has been modified as suggested by the reviewer. A12.2 is now colored in dark yellow, while Rrn11 is depicted in bright yellow. Stalk, heterodimer and A12.2 are now labeled in the cartoon presentation. The cleft closure is now also indicated and in the elongation complex RNA is depicted in red (upper panel right).

24. "Unusual DNA position in the Pol I PIC" - unusual with respect to what? Maybe better say 'Unexpected'.

Response: We changed the text in page 6 to *“Unexpected DNA position in the Pol I PIC”* as the reviewer suggested.

25. Page 6. What do the authors mean by fold recognition. This should be explained.

Response: We replaced “Fold recognition” by *“identification of macromolecular folds”* in the text in page 6 and the Figure EV4 legend.

26. What is the A135 'wedge'? If it is a loop, it should be named according to flanking secondary structure elements so that it can be easily identified.

Response: The A135 wedge corresponds to A135 residues 813-819. The term “wedge” was used by Barnes et al., 2015, where this element was seen stabilizing upstream DNA. We included residues numbers and the reference to the text on page 7.

27. First sentence of discussion: it is unclear what 'first' refers to, the PIC or the open state. This should be rephrased, now taking into account the Engel et al. paper.

Response: We rephrased the sentence to “Our Pol I PIC model shows major differences to the Pol II system and to previously proposed Pol I PIC models (Hoffmann et al, 2016; Knutson et al, 2014).”

Figure for reviewer. Densities from the Pol I PIC.

A Cryo-EM density for the DNA template strand in the open region with the fitted DNA (blue) compared to the DNA template strand in the EC (orange) indicating the different tilt of the DNA. **B** Density for the Rrn7 Zn ribbon is shown with its position overlapping with the one in the ITC. **C** Extra density appears at a lower threshold in a similar position as the A49 tWH in the Pol I ITC, although the missing linker in the cleft combined with low resolution precludes unambiguous positioning. The A49 tWH also needs to change its position compared to its position in the EC_tWH (Tafur et al, 2016, PDB ID 5m64) as this position clashes with the N-terminal moiety of Rrn3.

Thank you for submitting a revised version of your manuscript. It has now been seen by the original referee and this person's comments are shown below. As you will see the referee finds that all criticisms have been sufficiently addressed and recommends the manuscript for publication, pending minor textual clarification. I would therefore invite you to submit a final version that incorporates these minor changes and also address the following editorial points:

-> The crosslink data for figure EV4 will have to be relabeled as a source data file and a brief legend describing the content should be included a separate read me text file in the ZIP archive. However, for simplicity we can do this for you in-house by moving the already provided legend from the Appendix file to the Source data file. Could you let me know if you agree to this?

-> As a consequence of that change we will need you to provide a new Appendix file in which this legend has been removed. In addition, could you please change the title on the first page of the pdf to just say 'Appendix'?

-> We also noticed that callouts are currently missing for Fig EV4D and Fig EV5A, B and E. In addition, the callouts for Fig EV4A, B and C appear before Fig EV2A-C and EV3. Could you please provide these call-outs in the final version?

-> As a final, very minor point EV3 has only one figure but is called EV3A in the EV figure legend and in the figure itself.

Thank you again for giving us the chance to consider your manuscript for The EMBO Journal, I look forward to receiving your final revision.

 REFEREE REPORT

Referee #1:

The authors have strongly improved the manuscript and addressed our concerns. This reviewer highly recommends publication of this important contribution. I encourage the authors to submit their final model to the protein databank (PDB). Please note the below minor points. It is up to the authors whether they wish to consider this before publication.

1. Page 10: 'Interestingly, the C-terminal cyclin domain of Rrn7 also exposes its positive patch upstream to the DNA bend, likely stabilizing this DNA conformation by electrostatic attraction.' Which sequence position is this according to the CF crystal structure? Is there a direct DNA-contact to the C-terminal cyclin? This would contradict the author's previous statement on page 9 that "...the C-terminal cyclin domain of Rrn7 is not positioned as closely to the DNA as in Pol II, where the C-terminal cyclin domain of TFIIB contacts DNA ...".

2. Page 10: 'We were able to assign 10 TPR helices at the C-terminal end of Rrn11 into the cryo-EM density ...' Which are those 10 helices and can they be marked in a figure? Why do figure 3 and figure S3 only show 9 helices? How does this compare to the 9 TPR helices in the CF crystal structure, meaning which helix is additionally assigned?

3. The concept of the 'early PIC' was removed from the manuscript. Nevertheless, construction of an 'early PIC model' is still described in the methods and should be removed.

4. Note the reference Engel et al. 2013 is correctly referred to in the text but not present in the reference list.

Point-by-Point Response

1. Editorial Points

The crosslink data for figure EV4 will have to be relabeled as a source data file and a brief legend describing the content should be included in a separate read me text file in the ZIP archive.

However, for simplicity we can do this for you in-house by moving the already provided legend from the Appendix file to the Source data file. Could you let me know if you agree to this?

Response: We would be grateful if you could do this for us!

As a consequence of that change we will need you to provide a new Appendix file in which this legend has been removed. In addition, could you please change the title on the first page of the pdf to just say 'Appendix'?

Response: We will provide a new Appendix file with removed legend and changed title.

We also noticed that callouts are currently missing for Fig EV4D and Fig EV5A, B and E. In addition, the callouts for Fig EV4A, B and C appear before Fig EV2A-C and EV3. Could you please provide these call-outs in the final version?

Response: We included callouts for Fig EV4D (p. 20) and Fig EV5A, B (p. 11) and E (p. 12). In addition we moved the callouts for Fig EV4A, B and C (p.10) that now appear after the callouts for Fig EV2A-C and EV3.

As a final, very minor point EV3 has only one figure but is called EV3A in the EV figure legend and in the figure itself.

Response: In EV3 the "A" has been removed in the EV figure legend and in the figure itself.

2. Remaining Points of the Reviewer

Reviewer: I encourage the authors to submit their final model to the protein databank (PDB).

Response: Following the suggestion of the referee we have now submitted the coordinates to the PDB.

Minor points

1. Page 10: 'Interestingly, the C-terminal cyclin domain of Rrn7 also exposes its positive patch upstream to the DNA bend, likely stabilizing this DNA conformation by electrostatic attraction.' Which sequence position is this according to the CF crystal structure? Is there a direct DNA-contact to the C-terminal cyclin? This would contradict the author's previous statement on page 9 that "...the C-terminal cyclin domain of Rrn7 is not positioned as closely to the DNA as in Pol II, where the C-terminal cyclin domain of TFIIB contacts DNA ...".

Response: In our model the last helix in the C-terminal cyclin fold lies in close vicinity to the upstream DNA without directly contacting it. In our model and the CF crystal structure the last helix of the C-terminal cyclin fold corresponds to residues 494-514. To better describe this situation we now state in the manuscript, page 10: "*Interestingly, the C-terminal cyclin domain of Rrn7 also exposes a positive patch mainly formed by its last helix α 12 towards upstream DNA (Fig. 3, Fig EV3), likely contributing to stabilize this DNA conformation by electrostatic attraction without directly contacting the DNA.*"

2. Page 10: 'We were able to assign 10 TPR helices at the C-terminal end of Rrn1 1 into the cryo-EM density ...' Which are those 10 helices and can they be marked in a figure? Why do figure 3 and figure S3 only show 9 helices? How does this compare to the 9 TPR helices in the CF crystal structure, meaning which helix is additionally assigned?

Response: We only assigned 9 TPR helices as also depicted in our figures and observed in the crystal structure. Our assigned helices correspond to residues 208-440 in the CF crystal structure. We apologize for this error. In the method section we correctly state that 9 helices were assigned to Rrn11 (p. 21).

3. The concept of the 'early PIC' was removed from the manuscript. Nevertheless, construction of an 'early PIC model' is still described in the methods and should be removed.

Response: We removed this part from the method section.

4. Note the reference Engel et al. 2013 is correctly referred to in the text but not present in the reference list.

Response: We have added Engel et al., 2013 to the reference list.

3rd Editorial Decision

26 June 2017

Thank you for submitting the final revision of your manuscript to The EMBO Journal, I am pleased to inform you that your study has now been officially accepted for publication here.

Corresponding Author Name: Christoph W. Muller

Manuscript Number: EMBOJ-2017-96958